# In Vitro and In Vivo Comparison of Bone Growth Characteristics in Additive-Manufactured Porous Titanium, Nonporous Titanium, and Porous Tantalum Interbody Cages

**DOI:** 10.3390/ma15103670

**Published:** 2022-05-20

**Authors:** Meng-Huang Wu, Ming-Hsueh Lee, Christopher Wu, Pei-I Tsai, Wei-Bin Hsu, Shin-I Huang, Tzu-Hung Lin, Kuo-Yi Yang, Chih-Yu Chen, Shih-Hao Chen, Ching-Yu Lee, Tsung-Jen Huang, Fang-Hei Tsau, Yen-Yao Li

**Affiliations:** 1Department of Orthopaedics, School of Medicine, College of Medicine, Taipei Medical University, Taipei 110301, Taiwan; maxwutmu@gmail.com (M.-H.W.); aleckc2424@gmail.com (C.-Y.C.); ejaca22@gmail.com (C.-Y.L.); tjdhuang@gmail.com (T.-J.H.); 2Department of Orthopedics, Taipei Medical University Hospital, Taipei 110301, Taiwan; 3TMU Biodesign Center, Taipei Medical University, Taipei 110301, Taiwan; 4Neurosurgery, Department of Surgery, Chang Gung Memorial Hospital, Chiayi 613016, Taiwan; ma2072@gmail.com; 5Department of Nursing, Chang Gung University of Science and Technology, Chiayi 613016, Taiwan; 6College of Medicine, Taipei Medical University, Taipei 110301, Taiwan; cjwuchris@gmail.com; 7Biomedical Technology and Device Research Laboratories, Industrial Technology Research Institute, Chutung, Hsinchu County 310401, Taiwan; peiyi@itri.org.tw (P.-I.T.); sophiaShinI@itri.org.tw (S.-I.H.); yangkuoyi@itri.org.tw (K.-Y.Y.); 8Sports Medicine Center, Chang Gung Memorial Hospital, Chiayi 613016, Taiwan; weibinhsu@gmail.com; 9Material and Chemical Research Laboratories, Industrial Technology Research Institute, Chutung, Hsinchu County 310401, Taiwan; dustyLin@itri.org.tw; 10Department of Orthopedics, Shuang-Ho Hospital, Taipei Medical University, Taipei 235041, Taiwan; 11Department of Orthopedic Surgery, Buddhist Tzu-Chi General Hospital, Taichung Branch, Taichung 427213, Taiwan; shihhaotzuchi@gmail.com; 12Department of Orthopedic Surgery, Tzu-Chi University, Hualien 970374, Taiwan; 13Laser and Additive Manufacturing Technology Center, Southern Region Campus, Industrial Technology Research Institute, Tainan 734045, Taiwan; fanghei@itri.org.tw; 14Department of Orthopaedic Surgery, Chang Gung Memorial Hospital, Chiayi 613016, Taiwan; 15College of Medicine, Chang Gung University, Taoyuan 333323, Taiwan

**Keywords:** interbody fusion, porous titanium, spinal fusion, three-dimensional printing, selective laser melting

## Abstract

Autogenous bone grafts are the gold standard for interbody fusion implant materials; however, they have several disadvantages. Tantalum (Ta) and titanium (Ti) are ideal materials for interbody cages because of their biocompatibility, particularly when they are incorporated into a three-dimensional (3D) porous structure. We conducted an in vitro investigation of the cell attachment and osteogenic markers of self-fabricated uniform porous Ti (20%, 40%, 60%, and 80%), nonporous Ti, and porous Ta cages (*n* = 6) in each group. Cell attachment, osteogenic markers, and alkaline phosphatase (ALP) were measured. An in vivo study was performed using a pig-posterior-instrumented anterior interbody fusion model to compare the porous Ti (60%), nonporous Ti, and porous Ta interbody cages in 12 pigs. Implant migration and subsidence, determined using plain radiographs, were recorded before surgery, immediately after surgery, and at 1, 3, and 6 months after surgery. Harvested implants were assessed for bone ingrowth and attachment. Relative to the 20% and 40% porous Ti cages, the 60% and 80% cages achieved superior cellular migration into cage pores. Among the cages, osteogenic marker and ALP activity levels were the highest in the 60% porous Ti cage, osteocalcin expression was the highest in the nonporous Ti cage, and the 60% porous Ti cage exhibited the lowest subsidence. In conclusion, the designed porous Ti cage is biocompatible and suitable for lumbar interbody fusion surgery and exhibits faster fusion with less subsidence compared with porous Ta and nonporous Ti cages.

## 1. Introduction

Interbody cages have been increasingly used as an adjunct in interbody arthrodesis during spinal surgery to treat degenerative disorders of the lumbar spine [1]. Interbody fusion enables a higher fusion rate than posterolateral fusion when applied in various approaches (e.g., anterior, posterior, transforaminal, extreme, or direct lateral) and for oblique lumbar interbody fusion. Interbody fusion considerably reduces the rate of reoperation attributable to pseudarthrosis, implant failure, or donor graft site problems [2,3,4]. Fusion rates of as high as 96% and 100% have been reported with standalone cages and cages with additional posterior instrumentation, respectively, at 2-year postoperative follow-up [5]. The primary goals of using interbody support are to correct existing mechanical deformations and to provide primary stability to the motion segment until interbody fusion is achieved [6]. Autogenous bone grafts are frequently used for interbody fusion. They possess osteogenic, osteoinductive, and osteoconductive properties, and they are not associated with disease transmission risk. However, they have several limitations, including a long surgery duration, donor graft site pain, pseudarthrosis, limited bone sources, and graft retropulsion or collapse risk [7]. Allogenic bone grafts are an alternative to autogenous bone grafts; however, up to 40% resorption may occur during the early postoperative period when they are used, leading to kyphotic deformity and residual neural structure compression [8]. Interbody cages were developed to overcome the disadvantages associated with interbody fusion conducted with autogenous or allogenic bone grafts [9]. However, most cage materials, especially stainless steel (stiffness, 200 GPa), titanium (Ti, stiffness, 110 GPa), porous tantalum (PTa; stiffness, 3 GPa), carbon fiber (stiffness, 3–7 GPa), and polyetheretherketone (PEEK; stiffness, 3.6 GPa), exhibit higher stiffness than the spinal motion segment, vertebrae bones (2.1 GPa) and cortical bones (2.4 GPa) [10,11,12]. In addition to stiffness, the hydrophilicity, contact angle, and biocompatibility of the cage material are key factors that influence bone healing. Poor bone integration properties may lead to stress shielding, subsidence, cage migration, or pseudarthrosis of the motion segment [13].

An ideal bone substitute is a porous implant material with a favorable pore structure and mechanical properties [14]. Interconnected pores allow tissue ingrowth such that the prosthesis can be anchored to the surrounding bone to promote implant longevity. When a porous implant is inserted into the bone marrow, fibrin and fibrous tissue and subsequently cancellous bone are formed. The cancellous bone is remodeled into a lamellar bone, and bone-marrow-like tissue replaces the fibrous tissue. Differentiation progresses from the opening to the center of the implanted material, and the movement speed of the marrow front is estimated to be 3–4 mm every 6 weeks [15,16]. Low stiffness and hardness can be achieved through a porous implant design [16]. PTa, which has satisfactory bone growth properties, is the most used material for constructing porous metal cages for interbody fusion [12,17].

Rapid prototyping, which was first developed in the 1980s, is a revolutionary manufacturing method for rapidly fabricating objects of various shapes. Three-dimensional (3D) objects can be generated automatically with this method by using computer-aided manufacturing and computer-aided design (CAD) data, which have been employed to fabricate scaffolds. The lattice structures of orthopedic implants can be easily produced through additive manufacturing (AM). These lattice structures exhibit anisotropic mechanical properties that resemble those of human bone [18,19]. Selective laser melting (SLM) is a powder-based AM technique in which objects are fabricated from a 3D CAD model by using a high-energy laser beam that fuses metal powders in its focal zone layer by layer. SLM is used to manufacture implants that are similar in structure to human cancellous bone; notably, such implants have yet to be produced using pure Ti. In one study, porous Ti (PTi) implants were found to be effective bone substitutes in vivo, as indicated by the osteoinductivity and osteoconductivity induced by chemical and thermal treatment [14]. An increase in pore size results in an increase in permeability, and high-permeability implants possess osteoconductive potential. Implants with the optimal pore size (approximately 500 μm) can induce the growth of directly mineralized bones and enhance capillary formation to promote bone formation [20,21]. Moreover, a controllable pore size can result in improved vascularization and higher amounts of nutrients in the interior of a PTi implant. Hui et al. observed that high implant permeability enhances healing and integration at the bone graft–host interface [22]; they also reported that the lowest threshold of conductance through cancellous bone for vascularization and the formation of osteoblasts was 1.5 × 10^−10^ m^3^s^−1^ Pa^−1^. Animal and human studies have tested various additive cages and reported satisfactory results [23]. The PTi implants produced through SLM have a similar compressive strength to those produced using PTa. Moreover, they have homogenous porosity, which promotes bone ingrowth. Therefore, SLM is a promising technique for fabricating interbody cages. To investigate the mechanical properties and bone ingrowth characteristics of PTi cages manufactured through SLM, we evaluated and compared the structure, mechanical properties, and biocompatibility of PTi, nonporous Ti (NPTi), and PTa interbody cages in vitro, as well as the bone ingrowth characteristics of these cages in vivo.

## 2. Materials and Methods

### 2.1. Preparation and Microstructures of Pti Interbody Cages

Five types of scaffold Ti-6Al-4V 3D structures, with porosity levels of 0%, 20%, 40%, 60%, and 80%, were fabricated through SLM (EOSINT M270, EOS GmbH, Munich, Germany; Figure 1). The pore design of the cages was determined using numerical and experimental approaches [24], and the shape of the fabricated PTi cages was based on that of a fabricated PTa cage (60–80% porosity; TM-S, 11 × 14; angled height, 5 mm; Zimmer, Warsaw, IN, USA) for further comparison. The PTi cages were fabricated with an SLM machine by using pure Ti-6Al-4V powder (purity: >99.5%). The pore size of the PTi cages was examined through scanning electron microscopy (SEM; DSM940 Zeiss model, Carl-Zeiss, Ober-kochen, Germany), and cage surface area was calculated using SolidWorks (SolidWorks, Waltham, MA, USA).

### 2.2. Compression Test for Each Cage Type

Compression strength was evaluated using a strength testing device (Instron 3382, Instron, Norwood, MA, USA) with an axial load cell of 100 kN. The compression test and equipment were conducted and used, respectively, in accordance with the regulatory standards of ASTM International. Loading was applied at a rate of 0.2 mm/s under displacement control until a displacement of 1.5 mm was achieved. A load–displacement curve was plotted using Bluehill LE (Instron); yield displacement (mm), yield load (N), stiffness (N/mm), ultimate displacement (mm), and ultimate load (N) were recorded. The failure mode and load corresponding to a displacement of 1 mm (*n* = 5 for each of the examined three groups) were also recorded.

### 2.3. Pullout Test for Each Cage Type

A pullout test (speed 5 mm/min) was conducted for each cage type by using an Instron-4467 Tensile Tester (Instron), and pullout force was measured using an Instron 5-kN static load cell (Instron). During the pullout tests, the cages were fixed between two 320.37-kg/m^3^ solid rigid polyurethane foams that were anchored using L-shaped aluminum fixtures. A 250-N force, as measured using FlexiForce force sensors (Tekscan, Boston, MA, USA), was applied to the L-shaped fixtures to secure the foams and cage. The foams were used to simulate a consistent and uniform material with properties similar to those of human bone. A 250-N force was applied to both sides of each cage (*n* = 4 for each group).

### 2.4. Biocompatibility Tests for Each Cage Type: Osteoblast Adhesion and Differentiation of PTi Cages

To conduct an in vitro cell culture analysis, 2 × 10^5^/50 μL MG-63 cells (*Homo sapiens* bone osteosarcoma cell line; CRL-1427, American Type Culture Collection) were seeded onto NPTi and PTi cages (with porosity levels of 20%, 40%, 60%, and 80%) and PTa cages (*n* = 6 for each group). The cells were then cultured in a growth medium (α-minimum essential medium + 10% fetal bovine serum) for 4 days. For mineralization, the cells were treated with β-glycophosphate and dexamethasone for 18 days. Alkaline phosphatase (ALP) activity was analyzed to assess cell differentiation by using an ALP assay kit (Abcam, Cambridge, MA, USA). ALP mRNA expression and osteocalcin mRNA expression (a cell differentiation marker) were assayed on the final day of the experiment conducted in the present study. Total RNA was collected using TRIzol reagent (Thermo Fisher Scientific, Waltham, MA, USA) and subsequently subjected to complementary DNA (cDNA) synthesis with SuperScript IV Reverse Transcriptase (Invitrogen, Waltham, MA, USA). The synthesized cDNA (500 ng) was subjected to quantitative reverse transcription polymerase chain reaction (PCR), which was performed using a Bio-Rad CFX96 Real-Time PCR System with SYBR Green PCR Master Mix (Bio-Rad Laboratories, Hercules, CA, USA) and 10 nm sequence-specific primers. The oligonucleotides used for the PCR were 5′-CCACGTCTTCACATTTGGTG-3′ (forward primer) and 5′-AGACTGCGCCTAGTAGTTGT-3′ (reverse primer) for human ALP mRNA; 5′-ACCACAGTCCATGCCATCAC-3′ (forward primer) and 5′-TCCACCACCCTGTTGCTTGTA-3′ (reverse primer) for human glyceraldehyde-3-phosphate dehydrogenase; and 5′-CCCAAAGGCTTCTTGTTG-3′ (forward primer), 5′-CTGGTAGTTGTTGTGAGCAT-3′ (forward primer), 5′-ATGAGAGCCCTCACACTCCTC-3′ (forward primer), and 5′-GCCGTAGAAGCGCCGATAGGC-3′ (reverse primer) for osteocalcin [25]. Glyceraldehyde-3-phosphate dehydrogenase expression was used to normalize the abundance of the test RNA. Each detected gene was assayed in triplicate. For each specimen, cell attachment was evaluated through SEM examinations; for SEM, MG-63 cells (ATCC CRL-1427TM), a human bone osteosarcoma cell line, were incubated in minimum essential medium supplemented with 10% fetal bovine serum and cages (2 × 10^5^/50 μL for each implant) at 37 °C in 5% CO**_2_** for 14 days; thereafter, SEM (SolidWorks 2014, Waltham, MA, USA) examination was performed. All cages were first mounted on specimen stubs and sputter coated with 12 nm AuPd.

### 2.5. In Vivo Effects of PTi, NPTi, and PTa Cages in Porcine Lumbar Interbody Fusion Model: Animals and Design

An animal study was conducted in the present study to examine the in vivo effects of PTi, NPTi, and PTa cages in a porcine lumbar interbody fusion model. The animal study protocol was approved by the Ethics Committee of the Biomedical Technology and Device Research Laboratories of Taiwan’s Industrial Technology Research Institute. The protocol conformed to the guidelines of the Institutional Animal Care and Use Committee (approval No. 2013121916 and PIG-1040106) and the National Institute of Health for the use of laboratory animals. The animal study was conducted in accordance with the methods developed by Zou et al.; twelve 5-month-old female pigs (Pigmodel Animal technology Co, Lanyu, Taiwan), which weighed between 35 and 45 kg and were bred for scientific purposes, were selected from multiple litters [26] and handled in accordance with the regulations of the Institutional Animal Care and Use Committee for animal experimentation. Each pig underwent an anterior intervertebral lumbar arthrodesis at three levels, namely L1–L2, L3–L4, and L5–L6. Each level was randomly implanted with one of the three types of interbody fusion cages that were being tested. All the cages were filled with autologous iliac crest bone grafts, and each implant was secured with pedicle screws (Wiltrom, Hsinchu, Taiwan) through a posterior approach. All the pigs were kept in a single cage and fed a normal diet in which calcium and phosphorus accounted for 1.4% and 0.7%, respectively, of food weight. The pigs were observed for 6 months on average before they were euthanized.

### 2.6. Anesthesia

The animals were given intramuscular injections of a mixture of 5 mg/kg Zoletil 50 and 2.2 mg/kg xylazine and subjected to orotracheal intubation. Anesthesia was maintained through the inhalation of isoflurane (1.5%). A 1-gram dose of cephalosporin, a preoperative prophylactic antibiotic, was administered intravenously 30 min before surgery.

### 2.7. Surgery

Under aseptic conditions, the facet joints of the neighboring vertebrae were exposed through a posterior midline incision and paraspinal approaches. Pedicle screws with a 5 mm diameter and 25 mm length were transpedicularly inserted into the neighboring vertebrae and bilaterally connected to connecting rods (5 cm in length) between the second and third, fourth and fifth, and sixth (i.e., L1/L2, L3/L4, and L5/L6, respectively). Subsequently, the incision on a pig’s back was carefully sutured and closed. After left decubitus positioning was completed, a retroperitoneal anterior approach was employed. Under fluoroscopic control, the intervertebral spaces at the implanted levels were identified before surgical intervention. The rectus abdominis muscle and its sheath were incised and retracted. The innermost layer (i.e., the fascia of the transverse abdominis) was carefully dissected to prevent damage to the peritoneum, which lies immediately underneath. After the peritoneum and its contents were separated and retracted, the quadratus lumborum and psoas major muscles were visible. The anterior lumbar spine was easily identified because of its thick and shiny anterior longitudinal ligament. After the anterior and posterior borders of the vertebral discs were verified, the L1–L2, L3–L4, and L5–L6 intervertebral discs were excised with the cranial and caudal cartilage end plates, the ring apophysis, and a part of the anterior longitudinal ligament to form a gap of approximately 5 mm to fit the cages. A 5 mm trial cage was inserted into the disc space to confirm the positions of the fabricated cages through GE OEC Fluorostar 7900 fluoroscopy (GE Healthcare, Chicago, IL, United States). Autogenous bone grafts from the iliac crest and ribs were then morselized and packed into the central holes of the fabricated PTi, NPTi, and PTa cages. After the insertion of the three implants, the abdominal muscle and rectus abdominis sheath were carefully sutured and closed. The skin was then closed using running sutures. Before surgery, prophylactic cephalosporin (1 g) was intravenously injected; immediately after surgery, analgesic ketorolac (30 mg) was intramuscularly injected. All the pigs were kept in individual pens and fed a normal diet in which calcium and phosphorus accounted for 1.4% and 0.7%, respectively, of food weight. Ibuprofen (400 mg/tablet, two tablets/day) was administered to achieve analgesia for 7 days and then continued, if necessary. Seventeen days before euthanasia, 20 mg/kg tetracycline hydrochloride (T3383, Sigma-Aldrich, St. Louis, MO, USA) was intravascularly injected. The pigs were euthanized under deep anesthesia and then given an intravenous injection of 1–2 mEq/kg KCl. Next, the entire lumbar spinal column (L1–L7) was removed en bloc, stripped of soft tissue, transported to a laboratory, and stored at −20 °C until examination.

### 2.8. Radiographic Examination

To evaluate the morphology of the implant site before euthanasia, we generated plain radiographs immediately and at 1, 3, and 6 months after surgery. To evaluate cage fusion, we measured the lengths of the radiolucent lines at the cranial and caudal vertebrae–implant/graft interfaces and the total lengths of these interfaces on lateral radiographs [26]. Radiographic grading was performed by dividing the length of the radiolucent lines by the total length of the vertebrae–implant/graft interface. The radiolucent lines at the vertebrae–implant/graft interfaces represented three states of healing and were categorized according to their relative percentages into the following categories to represent the three states: >50% radiolucent lines, ≤50% radiolucent lines, and absence of radiolucent lines. To evaluate cage subsidence, we measured the fusion segment lengths on lateral radiographs at each time point and adjusted the measured lengths on the basis of the upper vertebral height for natural bone growth (Figure 2). The length of the connecting rods was 5 cm, which was half the sum of the lengths of the bilateral rods (red and blue lines in Figure 2). The vertebral growth ratio (VGR) is the ratio of the upper adjacent vertebral body length (green line in Figure 2) to the lower adjacent vertebral body length (yellow line in Figure 2) at a given time point on the day of the operation. Furthermore, the fusion segment length is the ratio of the length of the fusion vertebrae and interbody cage (yellow line in Figure 2) to the VGR multiplied by 5 cm (i.e., the length of the connecting rod).

### 2.9. Histomorphometry

Eleven specimens were harvested from each group 6 months after surgery for histological analysis. The specimens were fixed in 10% formalin for several days, dehydrated sequentially with increasing concentrations of ethanol (70%, 95%, and 100%) for at least 1 day, and infiltrated for 5 days with polymethylmethacrylate (M55909-1L, Sigma-Aldrich). The specimens were cut horizontally and perpendicular to the axis of the bony end plates at the level of the bone–implant interface. Each specimen was cut longitudinally through the center into left and right halves by using a water-cooled diamond saw (Buehler slow speed saw model no. 11-1280-170, Lake Bluff, IL, USA). For each specimen, one half was randomly selected for histological processing, and the other half was used for a fluorochrome study. The specimen sections were cut to a thickness of approximately 150 μm by using an IsoMet Low Speed Saw (Buehler, Lake Bluff, IL, USA) and ground to a thickness of 60 μm by using a grinding and polishing machine (PATENT P20FR-HA, Taipei, Taiwan) (*n* = 6 for each group). The sections were then stained with Sanderson’s rapid bone stain (Dorn & Hart Microedge, Loxley, AL, USA) and counterstained with acid fuchsin. For qualitative histological analysis, all the sections were blindly evaluated by two reviewers, and all the bone–implant interfaces, bone tissue types, bone marrow, and fibrous tissue were identified through visible light microscopy (Olympus BX43, Tokyo, Japan). In addition to rapid bone scanning, the sections were examined through fluorescence microscopy (Olympus BX43 U-RFL, Tokyo, Japan) to identify the formation of new bone, which was labeled with tetracycline. Yellow fluorochrome labeling was used to indicate the lamellar bone identified under polarized light. Each histological section was defined as (1) a bone inside a cage between two end plates (bone graft), (2) a bone inside pores (bone ingrowth), or (c) a bone that surrounds an implant (bone ongrowth); the aforementioned sections are presented in Figure 3. Bone healing within Zone A (i.e., graft healing) was estimated on the basis of a modified bone healing scoring scale [27]. This scale was scored from 1 for fibrous tissue to 10 for mature bone (Table 1).

#### 2.9.1. Backscattered-Electron SEM

Uncalcified tissue was embedded using a Technovit 9100 system (Kulzer, Wehrheim, Germany) and then cut into slices with a thickness of 1 mm. These slices were carefully polished and coated with carbon for backscattered-electron SEM (BSE-SEM; *n* = 5 for each group). A DSM940 camera (Carl Zeiss AG, Oberkochen, Germany) was used to capture images of the slices, which were then merged using the Photoshop Creative Cloud software (Adobe, San Jose, CA, USA) and converted into grayscale images. Next, the two-dimensional analysis function in the CT analysis software (CTAn, Bruker Skyscan, Konitch, Belgium) was applied to the images. We defined the implant area (i.e., the bone ingrowth area) as the region of interest, expanded the images to approximately 500 μm, excluded the bone ingrowth area, and defined the surrounding area as the bone ongrowth area. Subsequently, the morphometric indices of the ingrowth and ongrowth areas were analyzed.

#### 2.9.2. Statistical Analysis

Analyses were performed using GraphPad Prism 5 (GraphPad, La Jolla, CA, USA). The experimental data, which were all obtained from more than three experiments, are presented as means ± standard deviations. Statistical significance was set at *p* < 0.05. The Wilcoxon rank-sum test and Fisher’s exact test were conducted for nonparametric analysis. Data from more than two groups were compared using a one-way analysis of variance, and Tukey’s post hoc test was performed for repeated measures analysis. Power was set at 0.8. Sample size calculations indicated that five in vitro studies and 11 in vivo studies were required for each group. The in vitro and in vivo requirements were based on our preliminary data and the bone healing data presented by Zou et al. [26], respectively.

## 3. Results

### 3.1. Porosity and Surface Area

The surface area of the NPTi cage was 631 mm^2^. The surface area and pore diameter, respectively, were 2097 mm^2^ and 115.42 ± 19.21 μm for the 20% PTi cage, 2392 mm^2^ and 281.17 ± 38.26 μm for the 40% PTi cage, 2406 mm^2^ and 431.56 ± 22.98 μm for the 60% PTi cage, and 2137 mm^2^ and 561.17 ± 15.04 μm for the 80% PTi cage. Among the cages, the 60% PTi cage had the optimal pore size (approximately 500 μm) and the largest surface area.

### 3.2. Mechanical Compression Test and Pullout Test

A mechanical compression test was conducted to assess the mechanical properties of the fabricated cages under pressure. For a displacement of 1 mm, the load, stiffness, and Young’s modulus of the NPTi cage were 58273.42 ± 2330.9 N, 60246.76 ± 2409.8 N/mm, and 3165.63 ± 221.6 MPa, respectively. For the PTa cage, its load, stiffness, and Young’s modulus were 8536.67 ± 768.3 N, 7080.14 ± 566.4 N/mm, and 442.16 ± 26.5 MPa, respectively. For the 60% PTi cage, its load, stiffness, and Young’s modulus were 15964.79 ± 1117.5 N, 18951.63 ± 1516.1 N/mm, and 846.45 ± 59.3 MPa, respectively. For the 80% PTi cage, its load, stiffness, and Young’s modulus were 6930.63 ± 623.8 N, 6930.63 ± 623.8 N/mm, and 346.48 ± 17.3 MPa, respectively. For the aforementioned displacements, the NPTi and 80% PTi cages exhibited the highest and lowest loads, respectively. The load of the 80% PTi cage was comparable to that of the PTa cage (Table 2 and Figure 4).

A pullout test was conducted for the NPTi, 60% PTi, 80% PTi, and PTa cages. Among the cages, the 80% PTi cage had the highest pullout strength of 449.7 ± 2.5 N; the 60% PTi and NPTi cages had comparable pullout strength levels of 306.7 ± 6.5 N and 303.9 ± 2.8 N, respectively; and the PTa cage had the lowest pullout strength of 236.9 ± 8.8 N (Figure 4d).

### 3.3. Osteogenesis-Related Markers and Cell Attachment Test

To assess the biocompatibility of the fabricated cages, human MG63 cells were cultured to analyze osteogenesis markers, namely ALP activity and osteocalcin mRNA expression. After the induction of osteocyte differentiation, a higher level of ALP activity was observed in the group with 60% PTi cages than in the other groups (Figure 5). ALP expression and activity, which increased in a porosity-dependent manner, were the highest in the 60% PTi cages, suggesting that the cells in the high-porosity cages were in the early osteoblast stage and undergoing extracellular matrix secretion and matrix maturation. Because immature osteoblasts differentiate into mature osteoblasts, they express ALP to produce a mature extracellular matrix. Upon the completion of matrix maturation, late osteoblasts (mineralization phase) secrete osteocalcin [28]. The highest level of osteocalcin mRNA expression was observed in the NPTi group, and osteocalcin mRNA expression decreased with an increase in porosity, indicating a negative correlation between osteoblast differentiation and porosity. The high ALP activity and low osteocalcin expression levels suggested that the matrix mineralization stage of the examined cells was delayed. This delay in mineralization could have prolonged the cell proliferation stage, resulting in more bone-forming cells being produced within an implant. Cellular morphology was observed through SEM (Figure 6). The 60% and 80% PTi cages exhibited more favorable results than the other cages for cell migration inside pores, cluster formation inside pores, and cell linkage. Compared with the other cages, the PTa cage had a flatter cell morphology, and its cells were less interconnected with its surface.

### 3.4. In Vivo Examination of PTi Cages in Porcine Lumbar Interbody Fusion Model

Our results revealed that the 60% PTi cage had the optimal mechanical properties, biocompatibility, and adherent properties in vitro. Therefore, we investigated the radiographic outcomes and bone growth properties of the 60% PTi, PTa, and NPTi cages by using a porcine lumbar interbody fusion model (Figure 7a). Among the 12 pigs in the porcine model, one pig expired early because of poor postoperative recovery at 1 month after surgery. No evident displacement or collapse of any type of interbody cage was noted during the 6-month postoperative period. The radiographic grades were higher for the PTi and PTa cages than for the NPTi cage (Figure 7b). At 6 months post-surgery, the PTi cage had the longest fusion segment, indicating that it had the lowest subsidence among the three cages (Figure 7c). At the same time, the fusion segments in the PTi, PTa, and NPTi cages were 99.8%, 99.3%, and 98.3%, respectively (*p* < 0.05), of their original lengths.

### 3.5. Bone Formation Rate in Various Cages

Bone formation characteristics at 6 months post-surgery were determined through a histomorphometric analysis. Daily bone growth was categorized into three zones, namely Zones A, B, and C, which represented the bone graft inside the cage, bone ingrowth, and bone ongrowth, respectively (Figure 3 and Figure 8). For the NPTi cage, the average bone growth rate in Zone A was 0.58 ± 0.201 μm/day, and the average bone ongrowth rate in Zone C was 1.23 ± 0.52 μm/day. For the PTi cage, the average bone ingrowth rate in Zone B was 0.86 ± 0.35 μm/day, the average bone graft growth rate in Zone A was 1.04 ± 0.42 μm/day, and the average bone ongrowth rate in Zone C was 1.21 ± 0.39 μm/day. For the PTa cage, the average bone ingrowth rate was 0.86 ± 0.30 μm/day, the average bone graft growth rate was 0.97 ± 0.37 μm/day, and the average bone ongrowth rate was 0.97 ± 0.34 μm/day. Relative to the NPTi and PTa cages, the PTi cage had a significantly higher bone growth rate in Zone A (*p* < 0.05) but comparable bone growth rates in Zones B and C.

### 3.6. BSE-SEM: Bone Percentage Inside Pores and at Implant Interface

Bone volume and tissue volume were calculated to assess bone ingrowth in cage pores and bone ongrowth around the cages (Figure 9a–e). Among the cages, the percentage of bone ingrowth at 6 months post-surgery was the highest in the PTi cage (20.98% ± 4.8%) and significantly lower in the PTa cage (6.13% ± 3.2%) than in the PTi cage. The percentage of bone ingrowth was not calculated for the NPTi cage because it contained no pores. Among the cages, the percentage of bone ongrowth was the highest in the NPTi cage (48.85% ± 8.3%), followed by the PTi (36.02% ± 5.8%) and PTa (26.29% ± 7.8%) cages.

### 3.7. Bone Healing Score

Bone healing scores (ranging from 1 to 10) were assigned by two reviewers (Figure 10). The bone healing scores for the bone grafts inside the cages (Zone B) were 8.83 ± 0.75, 8.33 ± 1.21, and 5.17 ± 0.98 for the PTi, PTa, and NPTi cages, respectively. The PTi cage exhibited higher bone growth and contained more mature bone relative to the other cages; it also exhibited trabecular bone formation with complete bone bridging between implants. Some immature bone was observed in the PTa cage; however, it exhibited cartilage formation, which indicated a lower bone maturity relative to the PTi cage. Finally, the NPTi cage, which mostly contained cartilage with a small amount of immature bone, exhibited the lowest healing rate among the examined cages.

## 4. Discussion

With the increasing demand for spinal fusion surgery, the use of interbody cages has also increased [1]. Although interbody cages have yielded consistent results in spinal fusion surgery, current designs lack clinically favorable characteristics. To improve the fusion rate of interbody cages, various materials and designs aimed at increasing bone growth rates and accelerating recovery have been tested. Furthermore, various clinical risks (e.g., migration, subsidence, and pseudarthrosis) are associated with the use of interbody cages. These challenges can be overcome by using cage designs with suitable material properties [3,29]. Although porous cages have been designed to facilitate bone growth, their fabrication is affected by various manufacturing difficulties [30]. The powder sintering, combustion synthesis, and space holder methods cannot be employed to create pores with precise and consistent sizes. Furthermore, low pore connectivity is associated with several limitations, including those related to bone ingrowth and vascularization [31]. Porosity, pore size, and pore interconnectivity are key factors that considerably affect the mechanical properties and biological performance (including bone ingrowth, cell transportation, and nutrient transportation characteristics) of porous interbody cages [19]. However, the effects of these factors on cage performance are complex and even conflicting in specific situations. For example, an increase in porosity may be advantageous for biological processes but considerably reduce the stiffness and strength of a cage. Therefore, identifying the optimal conditions for porous interbody cages is a key task. In the present study, we investigated and compared the safety and biological effects of additive-manufactured PTi, NPTi, and PTa cages in vivo and in vitro. Our results indicated that of the three types of cages, PTi cages manufactured through SLM are the most biocompatible and suitable for use in lumbar fusion surgery because of their superior mechanical properties and bone growth characteristics relative to PTa and NPTi cages.

Ti is one of the most commonly used materials in AM because of its favorable biocompatibility in vivo, which can be attributed to its resistance to corrosion caused by bodily fluids, bioinertness, high fatigue limits, and osseointegration capacity [32]. In the presence of oxygen, Ti naturally forms a protective oxide film that enables it to withstand the harsh environment of the human body. Therefore, it has been used to fabricate interbody fusion cages [15,33]. In one study, a PTi cage without calcium phosphate, which had an interconnected porous structure, became osteoinductive when its bioactive surfaces were activated through chemical and thermal treatments [34]. Microrough surfaces, apatite-forming abilities, and complex porous structures are preconditions for osteoinduction. However, the optimal pore structure for interbody fusion has yet to be determined. Studies have used powder-sintered or plasma-sprayed porous bioactive Ti to evaluate the influence of porous structures on osteoinduction [15]. However, the conventional manufacturing methods employed in some studies did not provide precise control over the porosity, interconnectivity, pore size, and hardness of Ti alloys before the start of AM procedures [35,36]. Moreover, the high cost of Ti alloys limits their use. To overcome these disadvantages, considerable advances in 3D printing applications for AM technologies have been made in the past 5 years. These advances have enabled the production of fully functional Ti and Ti-alloy parts for biomedical applications [37]. The 3D printing method involves the fabrication of a 3D figure based on a digital design through AM and the continuous computer-controlled layering of materials [38]. Two-dimensional radiographic images (e.g., X-ray, magnetic resonance, and computerized tomography images) can be converted into 3D files to create customized anatomical and medical devices [39]. Powder metallurgy is a cost-effective method for fabricating Ti components, and powder bed fusion technologies can be employed to create hollow near-net shapes with high-resolution levels. Numerous studies have revealed that the mechanical properties of materials fabricated through AM are comparable or superior to those manufactured using conventionally fabricated Ti alloys [40]. Advances in the use of low-cost Ti powders are expected to expand the application of AM and reduce medical costs. The application of AM for cage production provides several advantages. The 3D printing method is highly cost-effective because it minimizes the use of unnecessary resources. Moreover, because materials are fabricated layer by layer in 3D printing, resources are not wasted and dropped powdered material can be recycled and reused. The 3D printing method is also faster than traditional manufacturing methods, which require time-consuming steps such as milling and forging. Moreover, the accuracy, reliability, and repeatability of 3D printing technologies allow for the fabrication of cages with consistent and accurate shapes such that more consistent increases in bone growth can be achieved [41]. The porous structure of Ti cages enhances bone ingrowth in lumbar interbody fusion. Furthermore, the design, hardness, and biocompatibility of interbody cages are crucial factors in lumbar interbody fusion. In the present study, the mechanical properties of the fabricated PTi cages were modified by adjusting their porosity to improve bone growth characteristics. At the time of writing, the characteristics of PTi cages have yet to be compared with those of cages made from other porous metal materials (e.g., PTa).

PTa is a material that is commonly used in clinical settings to manufacture interbody cages [42]. Tantalum (Ta, atomic number, 73; atomic weight, 180.05 u) is a transition metal that remains mostly inert in vivo [43]. PTa implants are fabricated through the pyrolysis of a thermosetting polymer foam, which results in the creation of a low-density vitreous carbon skeleton (98% porosity) with a repeating dodecahedron array of regular pores that resemble those of cancellous bone. Because of its structure, PTa has a high volumetric porosity, a low modulus of elasticity, and a high friction coefficient [44]. The components that are currently used in orthopedic applications have average pore sizes of between 400 and 600 mm and volume porosity levels of between 75% and 85% (99% Ta and 1% vitreous carbon by weight). A study reported that in addition to Ti, cobalt chromium (CoCr), stainless steel alloys, and other orthopedic biomaterials, PTa exhibits mechanical properties that are compatible with those of bone tissue [44]. In its solid form, Ta has a high modulus of elasticity (approximately 185 GPa). This high modulus of elasticity can be reduced considerably through porous carbon scaffolding, as was verified by a porous model [44], in which the modulus of elasticity was reduced to approximately 3 GPa. The modulus of elasticity of PTa resembles that of subchondral bone; however, the ultimate and yield strengths of PTa are 10 times greater than those of subchondral bone. Shimko et al. reported that the intrinsic permeability of PTa scaffolds ranges between 2.1 × 10^−10^ and 4.8 × 10^−1^ m^2^, which is similar to that of cancellous bone with the same level of porosity [20]. In the present study, the PTa cage exhibited favorable mechanical properties and bone growth characteristics, and its bone fusion properties were superior to those of the NPTi cage. PTa cages are an excellent option for clinical use; however, their availability is limited because of the scarcity of Ta. Relative to Ta, titanium is substantially more abundant and easier to obtain. Therefore, PTi cages may be more cost-effective to manufacture than PTa cages; however, this supposition requires further validation.

In the present study, the PTi cages manufactured through SLM had a porosity of between 60% and 80% and exhibited compressive strength that was similar to that of PTa cages. Moreover, all the PTi cages had a higher pullout strength relative to the PTa cages. Furthermore, the 60% PTi cage had higher levels of bone formation markers in vitro relative to the PTa cages. Among the examined cages, the levels of ALP expression and activity were highest in the 60% PTi cage, suggesting that the cells in high-porosity cages were less mature and in a proliferative state; by contrast, osteocalcin mRNA expression was highest in the NPTi cage and decreased when porosity increased, indicating that osteoblast differentiation decreased when porosity increased. The 60% PTi cage had low stiffness but high mechanical strength, and thus, it was the most favorable option among the examined cages for clinical use. An increase in porosity increased pullout strength and enhanced cell behaviors and bone ingrowth patterns. These findings are consistent with those of a study on sheep vertebral fusion that was conducted by Wu et al., who fabricated a PTi cage through electron beam melting and reported that this exhibited more favorable osseointegration and higher mechanical stability relative to a conventional PEEK cage [31].

The 60% PTi cage exhibited the least subsidence among the examined cages. This result may be attributed to the following factors. First, the PTi cage had a lower elastic modulus relative to the NPTi cage. Second, the PTi cage had the highest level of ALP activity among the examined cages. Higher pore interconnectivity was achieved with the PTi cage than with the other cages, enabling the formation of stronger and more mature bone because of the increased osseointegrative potential and bioactivity, which enhanced bone ingrowth and tissue differentiation [45]. The PTa cage had the second lowest subsidence because it contained mostly cartilage and immature bone. The osteoblast observed in the SEM micrograph of the PTa cage was considerably smaller and had lower interconnectivity than that observed in the SEM micrograph of the PTi cage. The NPTi cage exhibited the highest subsidence because its bone growth was relatively immature and mostly comprised of cartilage, which resulted in the formation of weak bone. The NPTi cage also exhibited the lowest level of ALP expression, which led to low bone growth. These results could be explained by the implant osseointegration in the PTi cage, which was the highest among the examined cages because that cage facilitated bone ingrowth and vascularization. This speculation is supported by a finding of an ovine study conducted by McGilvray et al. [46], who reported that PTi cages exhibited a considerably lower range of motion and a more favorable bone ingrowth profile relative to PEEK cages and plasma-sprayed PTi-coated PEEK cages.

Several studies investigated in vitro cell biocompatibility with various materials and porosity levels (e.g., Ta and Ti). A study conducted by Civantos et al. revealed that porous samples were superior to nonporous samples in promoting cell interactions and enabling adhesion with substrates; this was because their larger surface contact area (resulting from their larger pore size) promoted advanced adhesion to substrates, and they exhibited higher levels of cellular metabolic activity and ALP expression relative to fully dense substrates that promote osteogenesis [47]. Another study conducted by Balla et al. revealed that laser-processed PTa has properties similar to those of cortical bone. Furthermore, they demonstrated that living cell density and cell adhesion levels were significantly higher in PTA samples than in PTi samples [48]. However, their study only compared 27% PTa and 45% PTa samples with 27% PTi samples, and our study has revealed that 27% is not the optimal porosity for PTi samples.

Our study has several limitations. First, its follow-up duration was short. Second, the porcine model of our study differs from the model for human spinal fusion surgery. Therefore, future studies that involve longer follow-up durations and human clinical trials are warranted. Third, the pore design of the cages used in our study only had one configuration, and thus, future studies should explore other lattice designs [49]. Furthermore, researchers can also examine the effects of various AM methods (e.g., electron beam melting, laser sintering, and laser deposition), metal powder properties, and secondary finishing processes on the properties of fabricated interbody cages.

## 5. Conclusions

Our results indicate that the 60% PTi cage manufactured through SLM is biocompatible and suitable for use in lumbar fusion surgery. The mechanical properties and bone growth characteristics of this cage are superior to those of the other examined cages, namely the PTa and NPTi cages. The in vitro and in vivo characteristics of PTi cages resemble those of PTa cages, and they exhibit more favorable cell behavior and less subsidence and migration compared with NPTI cages.

## Figures and Tables

**Figure 1 materials-15-03670-f001:**
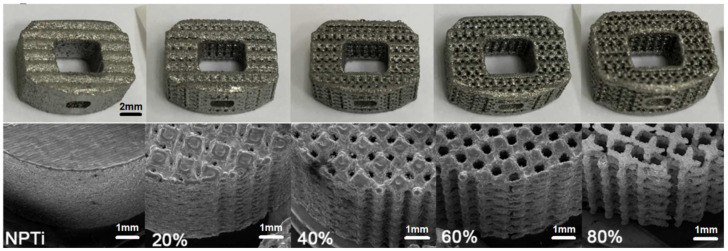
Additive-manufactured nonporous and porous cages (upper row, gross view; lower row, scanning electron microscopy 50×). NPTi, nonporous titanium cage; PTi, porous titanium cage.

**Figure 2 materials-15-03670-f002:**
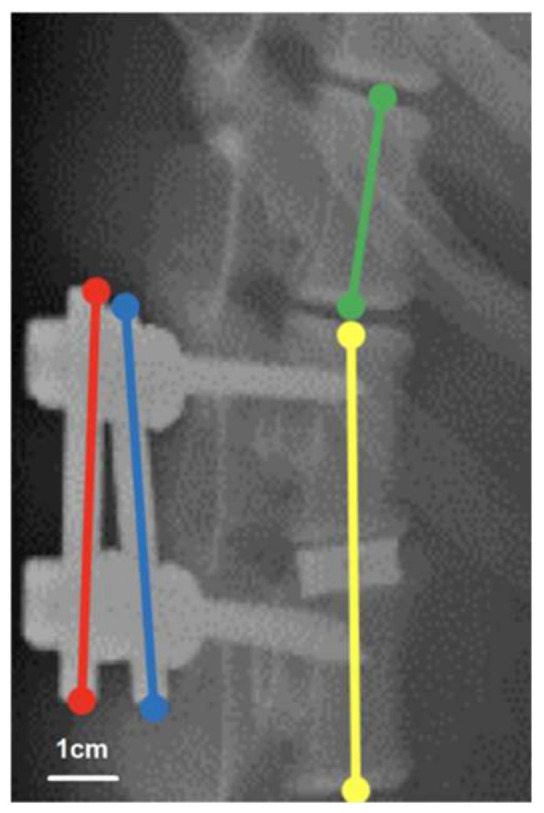
Assessment of fusion segment length. With a real connecting rod length of 5 cm, the sum of the measured length of the bilateral rods (red and blue line) divided by 2 is regarded as the measured connecting rod length, which should be 5 cm. For adjusting the bone growth of the vertebra, the vertebral growth ratio (VGR) is defined as the upper adjacent vertebral body length (green line) divided by the lower adjacent vertebral body length (yellow line) at a given time point on the day of the operation. The fusion segment length is defined as the length of the fusion vertebrae and interbody cage (yellow line) divided by the length of the measured connecting rod × 5 cm × VGR.

**Figure 3 materials-15-03670-f003:**
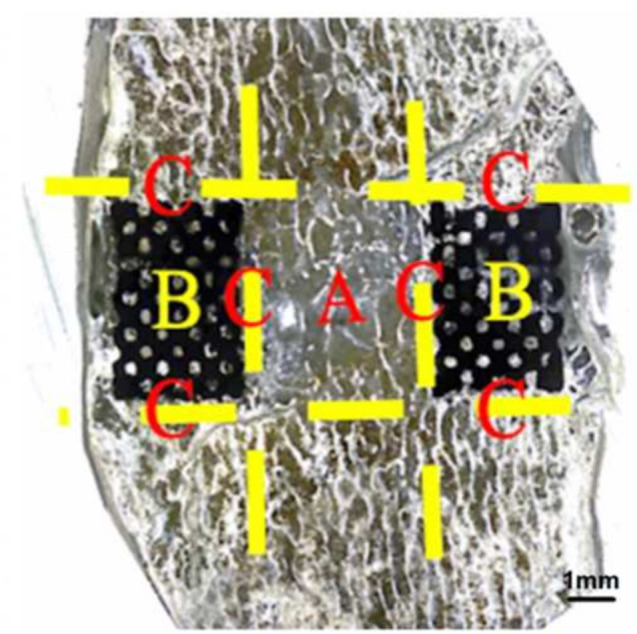
Assessment of bone ingrowth/ongrowth of porous cages on the histologic section. Zone A, bone inside cage between two endplates (Bone graft); Zone B, bone inside cage (bone ingrowth); Zone C, bone surrounding the implant (Bone ongrowth).

**Figure 4 materials-15-03670-f004:**
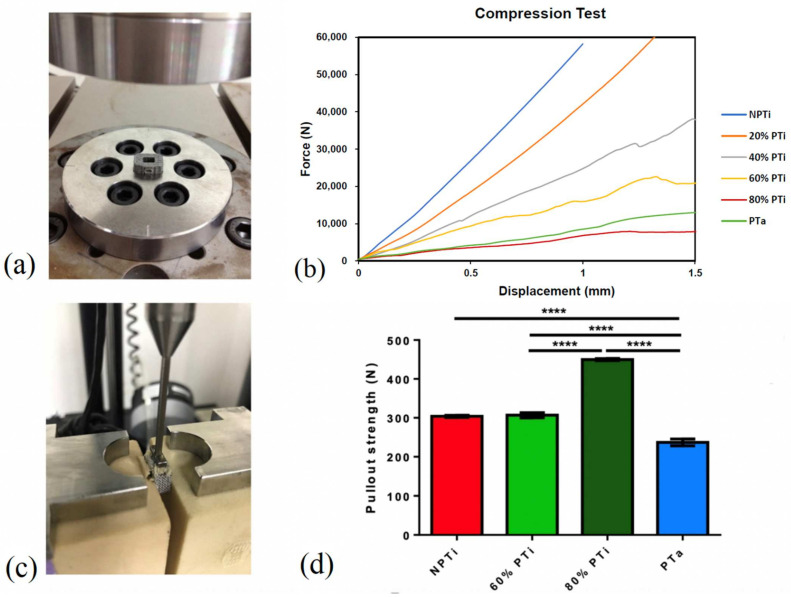
(**a**) Compression test model for nonporous titanium (NPTi), porous titanium (PTi) and porous tantalum (PTa) cages. (**b**) Compression test results for each cage, (*n* = 5). (**c**) Pullout test model for cages. (**d**) Pullout strength test results for NPTi, 60% PTi, 80% PTi, and PTa cages (*n* = 5). **** *p* < 0.0001.

**Figure 5 materials-15-03670-f005:**
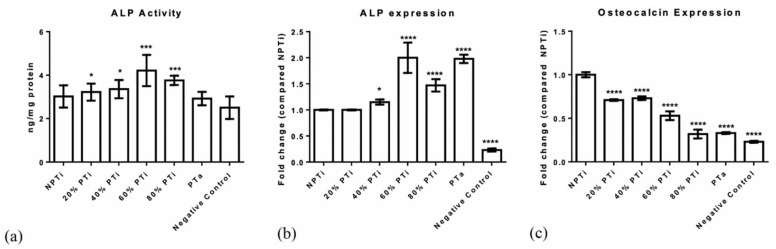
(**a**) Alkaline phosphatase (ALP) activity of nonporous titanium (NPTi), porous titanium (PTi), and porous tantalum (PTa) cages. (**b**) ALP mRNA expression of NPTi, PTi, and PTa cages. (**c**) Osteocalcin mRNA expression of NPTi, PTi, and PTa cages. *n* = 6; * *p* < 0.05, *** *p* < 0.001, **** *p* < 0.0001.

**Figure 6 materials-15-03670-f006:**
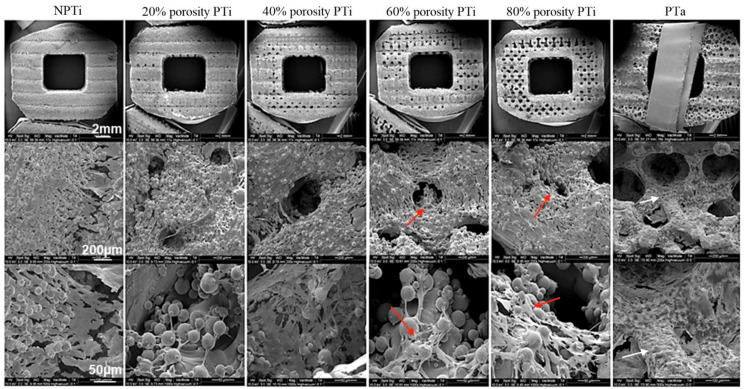
In vitro results for cell attachment. Scanning electron micrographs of nonporous titanium (NPTi), porous titanium (PTi), and porous tantalum (PTa) cages. Upper row, 1.7×; middle row, 200×; lower row, 500–1000×. Among the examined cages, 60% and 80% PTi cages exhibited more favorable results for cell migration inside pores (red arrows), cluster formation inside pores, and cell linkage; the PTa cage had a flatter cell morphology (white arrows), and cells were less interconnected on its surface.

**Figure 7 materials-15-03670-f007:**
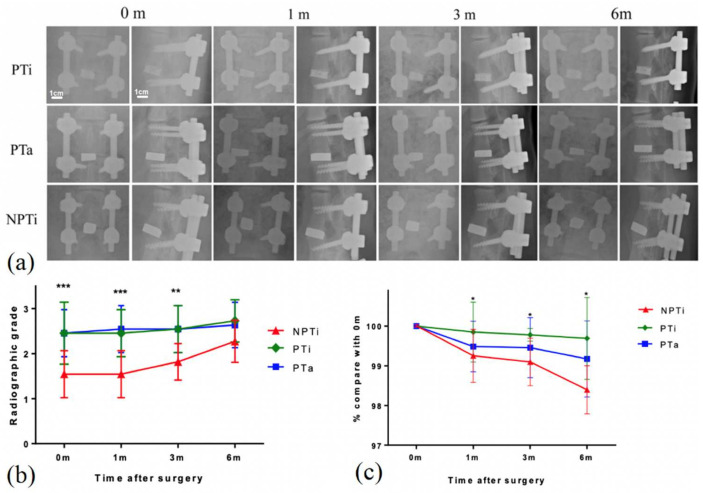
(**a**) Radiographic exam of every time point in porcine lumbar interbody fusion model of nonporous titanium (NPTi), 60% porous titanium (PTi), and porous tantalum (PTa) cages. (**b**) Radiographic grades and (**c**) fusion segment lengths of NPTi, PTi, and PTa cages at time of operation (op, 0 m), 1 month post-surgery (1 m), 3 months post-surgery (3 m), and 6 month post-surgery (6 m). * *p* < 0.05, ** *p* < 0.01, *** *p* < 0.001.

**Figure 8 materials-15-03670-f008:**
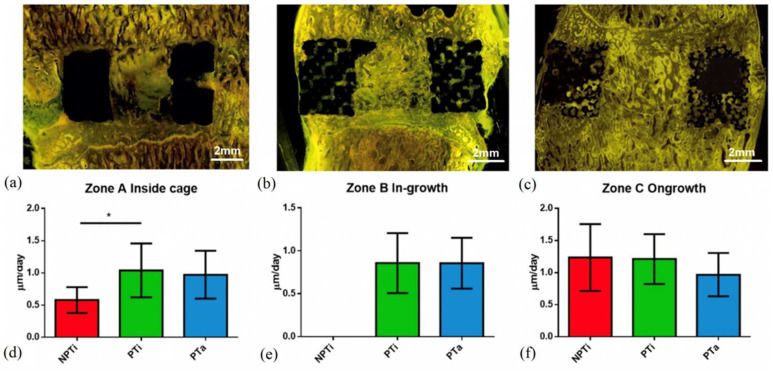
In vivo bone growth rate evaluated through fluorescence microscopy and labeled with tetracycline. (**a**) Nonporous titanium cage (NPTi); (**b**) 60% porous titanium cage (PTi); (**c**) porous tantalum cage (PTa); (**d**) A = bone inside cage between two endplates (bone graft), B = inside of cage (bone ingrowth), and C = bone surrounding the implant (bone ongrowth); (**e**) bone growth in Zone A; (**f**) bone growth in Zone B. (*n* = 6), * *p* < 0.05.

**Figure 9 materials-15-03670-f009:**
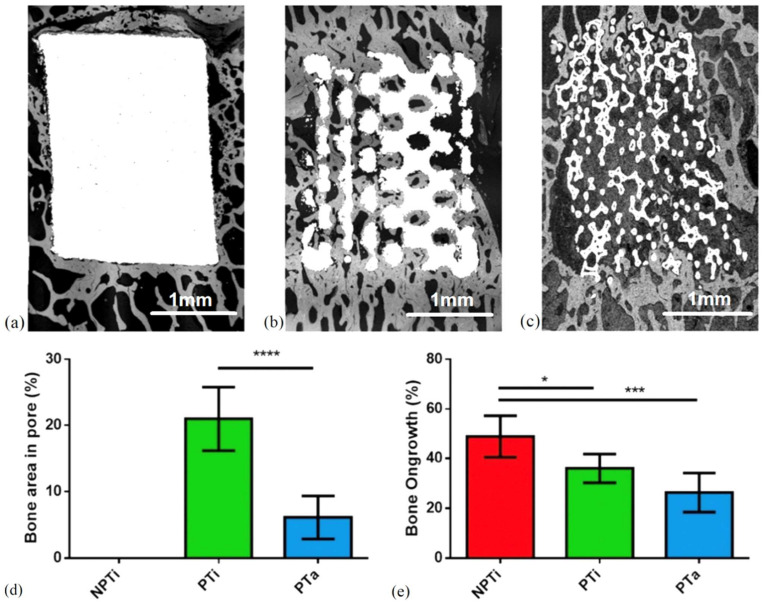
Backscattered-electron scanning electron micrographs displaying bone percentage for each cage type. (**a**) Nonporous titanium cage (NPTi), (**b**) 60% porous titanium cage (PTi), (**c**) porous tantalum cage (PTa), (**d**) bone area in cage pores, (**e**) bone ongrowth around cage. (*n* = 5), * *p* < 0.05, *** *p* < 0.001, **** *p* < 0.0001.

**Figure 10 materials-15-03670-f010:**
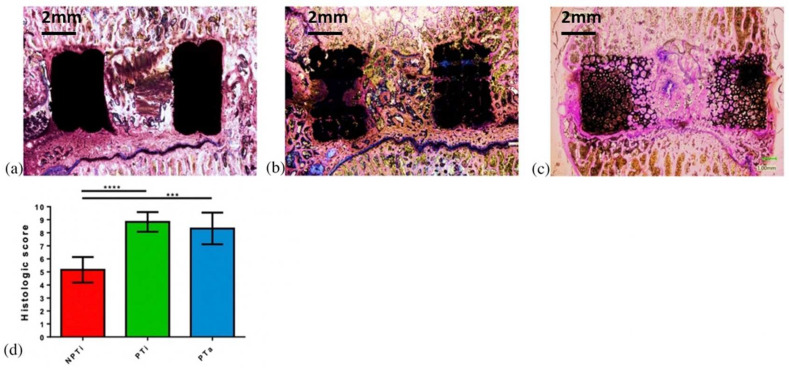
In vivo bone morphology stained with Sanderson’s rapid bone stain (Dorn & Hart Microedge, Loxley, AL) and counterstained with acid fuchsin. (**a**) Nonporous titanium cage (NPTi), (**b**) 60% porous titanium cage (PTi), and (**c**) porous tantalum cage (PTa); (**d**) modified bone healing scores for NPTi, PTi, and PTa. *n* = 6, *** *p* < 0.001, **** *p* < 0.0001.

**Table 1 materials-15-03670-t001:** Modified bone healing score for the healing of autogenous bone graft between two vertebral endplates derived from Perry et al. [27].

Score	Fibrous Tissue	Cartilage Tissue	Immature (Woven) Bone	Mature (Lamellar) Bone
1	100%	/	/	/
2	>50%	<50%	/	/
3	50%	50%	/	/
4	/	100%	/	/
5	/	>50%	<50%	/
6	/	50%	50%	/
7	/	<5%	>95%	/
8	/	/	100%	/
9	/	/	>50%	<50%
10	/	/	/	100%

**Table 2 materials-15-03670-t002:** Mechanical compression test of various porous cages.

Sample	Stiffness (N/mm)	1 mm Load (N)	Young’s Modulus (Mpa)
0% PTi	60,246.76 ± 2409.8	58,273.42 ± 2330.9	3165.63 ± 221.6
20% PTi	54,931.89 ± 2746.5	42,620.46 ± 2557.2	2285.14 ± 182.8
40% PTi	25,808.75 ± 1806.6	24,806.51 ± 1488.4	1333.38 ± 93.3
60% PTi	18,951.63 ± 1516.1	15,964.79 ± 1117.5	846.45 ± 59.3
80% PTi	6930.63 ± 623.8	6818.292 ± 545.5	346.48 ± 17.3
PTa	7080.14 ± 566.4	8536.67 ± 768.3	442.16 ± 26.5

PTa, porous tantalum cage; PTi, porous titanium cage (*n* = 5).

## Data Availability

Not applicable.

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
