# Peer review of "In Vitro and In Vivo Comparison of Bone Growth Characteristics in Additive-Manufactured Porous Titanium, Nonporous Titanium, and Porous Tantalum Interbody Cages"

_materials, 2022, doi:10.3390/ma15103670_

Round 1

Reviewer 1 Report

The authors aimed  to investigate the mechanical properties and bone ingrowth characteristics of PTi cages manufactured through SLM, we evaluated and compared the structure, mechanical properties,  and biocompatibility of PTi, nonporous Ti (NPTi), and PTa interbody cages in vitro as well as the bone ingrowth characteristics of these cages in vivo.

The study covers some issues that have been overlooked in other similar topics. The structure of the manuscript appears adequate and well divided in the sections. Moreover, the study is easy to follow, but few issues should be improved. Some of the comments that would improve the overall quality of the study are:

  1. Authors must pay attention to the technical terms acronyms they used in the text.
  2. English language needs to be revised.
  3. Conclusion Section: This paragraph required a general revision to eliminate redundant sentences and to add some "take-home message".

Reviewer 2 Report

In this manuscript, Wu et al. investigated and compared the mechanical characteristics, in vitro biocompatibility and in vivo bone regeneration potential of three different SML-manufactured cages (PTi, NPTi and PTa), with promising results for the 60% PTi cage, which exhibited a favourable biocompatibility, superior mechanical properties and bone in growth characteristics. However, despite the promising results that possess the potential of enriching the field of lumbar fusion surgery, the manuscript requires a few modifications ahead of publication.

Thus, the authors should consider the following suggestions:

1. I would recommend a revision of the written English throughout the whole length of the manuscript, due to some minor errors such as: “harvested implants were assed bone ingrowth and bone attachment (line 44)”; “Interbody fusion exhibits a higher fusion rate than does posterolateral fusion in various (line 56)”; “exhibit higher stiffness than do the spinal motion segment and vertebral and cortical bones (lines 75-76)”; “determined using a porous model in (???) [46], in which….” (line 529), etc.

2. Please abbreviate words such as “tantalum”; “titanium”; “alkaline phosphatase” at their first appearance in the manuscript (the abstract section) and use the abbreviation throughout the whole document.

3. Please remove the comma from the “in vivo” structure (line 40).

4. Please add the end bracket: “(n=6 in each group)” (line 39).

5. In the “Materials and Methods” section, subsection “2.4. Biocompatibility tests for each cage type: osteoblast adhesion, proliferation, and differentiation on PTi cages”, although the title implies the evaluation of the proliferative capacity of the newly designed scaffolds, the methodology for the aforementioned investigation does not appear in this subsection, which can be quite confusing for the readers. Was there any proliferation test performed? If no, then please modify the title of the subsection in order to reflect the actual investigations or if indeed the proliferation evaluation was performed, please add the method and the obtained results.

6. Moreover, I would recommend a more detailed methodology for SEM.

7. In subsection “3.3. Osteogenesis-related markers and cell attachment test” and “Discussion” section the authors wrote: “High ALP activity and low osteocalcin expression suggested that the cells were relatively young and were in a proliferative state”; respectively, “suggesting that cells in high-porosity cages are less mature and in a proliferative state.”, however no proliferation test was performed in order to sustain this affirmation. Moreover, the ALP evaluation was performed only at one experimental time point, therefore a time progression could not be observed.

8. Please add “cells/cm2” in line 157, after the cell density.

9. Please expand the legend of Figure 6 and explain shortly the changes in cell morphology visible on the micrographs and, if possible. I would suggest that the images should be accompanied by arrows that point the alterations/differences in morphology between the cells grown on different substrates.

10. It would be great if the authors could correlate their in vitro findings with similar data found in the specialized literature.

11. Please expand the abbreviation for “CoCr” (line 525).

Reviewer 3 Report

This is an excellent interdisciplinary paper consisting of material science, medical science, and biomedical engineering. A wide range of literature from each of the above perspectives was reviewed and the need/justification of this research is well established. The storyline flows naturally. 3 different materials were tested: PTi, NPTi, and PTa. For the PTi cages, 5 different porosities from 0%-80% were produced using SLM and tested for various properties such as tensile strength, pull-out strength, and finally implant. It was concluded that  the designed PTi cage is biocompatible and suitable for lumbar interbody fusion surgery, exhibiting faster fusion with less subsidence compared with PTa and NPTi cages. The conclusion was drawn based on experimental investigation. This paper definitely improves the current knowledge of the topic and will attract a range of readers and will finally attract citation. The manuscript deserves to be published. The followings are a few suggestions:

  1. Line 39: Please include the missing bracket ')'.
  2. Line 126: Please describe the method here. A manuscript should be standalone and should not be asked the reader to go through to another article to find the detailed experimental procedure. 
  3. Fig 4b: Please change the color of the legend. Especially, the difference between 60%PTi and PTa is not clear at all.
  4. A lot of scale bars were either missing or unclear in figures 1, 2, 3, 6, 7, 8, 9, and 10. please improve those qualities.
  5. Figure 6: need a better description. It is clear that there are 2 pictures for each time frame 0m, 1m, 3m and 6m. What is the difference between the two pictures for each time frame?
  6.  Figures 7-10: Please include which porosity of PTis are presented here in the legend/caption.
  7. The conclusion section is too small and the discussion section is too long. Please rearrange these sections and write a meaningful conclusion by outlining the true findings.

Round 2

Reviewer 2 Report

With the revision of manuscript, Wu et al. provided an improved version of their initial study. I am now generally in favor of publication; however, some minor corrections are still needed ahead of publication.

1. In the "Abstract" section "titanium" and "tantalum" should have been abbreviated simply as "Ti" and "Ta", respectively, while the PTi; NPTi; and PTa abbreviations should have been expanded to "porous Ti"; "nonporous Ti" and "porous Ta".

2. Please use the abbreviated form for "titanium" and "porous tantalum" in line 83, and remove PTi from between the brackets.

3. Please remove "porous titanium" from lime 116 and use the abbreviated form.

4. Please remove "nonporous Ti" from lime 135 and use the abbreviated form.

5. In the "Discussion" section please use the abbreviated form for "titanium" and "tantalum".

6. Please provide the modified version of Figure 6, so that the legend corresponds with the modified micrographs (no red or white arrows are visible).

Author Response

Response to Reviewer 2 Comments

With the revision of manuscript, Wu et al. provided an improved version of their initial study. I am now generally in favor of publication; however, some minor corrections are still needed ahead of publication.

Response: Thank you so much and we have made corrections according to your suggestion.

  1. In the "Abstract" section "titanium" and "tantalum" should have been abbreviated simply as "Ti" and "Ta", respectively, while the PTi; NPTi; and PTa abbreviations should have been expanded to "porous Ti"; "nonporous Ti" and "porous Ta".

Response 1: Thank you so much, we have abbreviated the “titanium” and “tantalum” to “Ti” and “Ta” while the PTi; NPTi, and PTa have been expanded in the abstract.

  1. Please use the abbreviated form for "titanium" and "porous tantalum" in line 83, and remove PTi from between the brackets.

Response 2: Thank you so much and we use the abbreviated form for “titanium” and “porous tantalum”, and remove PTi as your suggestion.

  1. Please remove "porous titanium" from lime 116 and use the abbreviated form.

Response 3: Thank you so much and we have removed “porous titanium” and use the abbreviated form as your suggestion.

  1. Please remove "nonporous Ti" from lime 135 and use the abbreviated form.

Response 4: Thank you so much and we have removed “nonporous Ti” and use the abbreviated form as your suggestion.

  1. In the "Discussion" section please use the abbreviated form for "titanium" and "tantalum".

Response 5: Thank you so much and we have made the changes as your suggestion.

  1. Please provide the modified version of Figure 6, so that the legend corresponds with the modified micrographs (no red or white arrows are visible).

Response 6: Thank you so much and we have provided the modified version. We are very sorry for providing the previous version and we are very grateful for your kind reminder.

Reviewer 3 Report

The authors made significant effort to address the reviewers’ comments. It can be accepted now.

Author Response

The authors made significant effort to address the reviewers’ comments. It can be accepted now.

Response: Thank you so much for your comment.